# A New Methodology for a Retrofitted Self-tuned Controller with Open-Source FPGA

**DOI:** 10.3390/s20216155

**Published:** 2020-10-29

**Authors:** Edson E. Cruz-Miguel, José R. García-Martínez, Juvenal Rodríguez-Reséndiz, Roberto V. Carrillo-Serrano

**Affiliations:** División de Investigación y Posgrado, Facultad de Ingeniería, Universidad Autónoma de Querétaro, Querétaro 76010, Mexico; ecruz30@alumnos.uaq.mx (E.E.C.-M.); jose.gm@uaq.mx (J.R.G.-M.); roberto.carrillo@uaq.mx (R.V.C.-S.)

**Keywords:** retrofitted, open-hardware FPGA, vibration analysis, genetic algorithm, instrumentation and sensors, adaptive and predictive control, controller self-tuning

## Abstract

Servo systems are feedback control systems characterized by position, speed, and/or acceleration outputs. Nowadays, industrial advances make the electronic stages in these systems obsolete compared to the mechanical elements, which generates a recurring problem in technological, commercial and industrial applications. This article presents a methodology for the development of an open-architecture controller that is based on reconfigurable hardware under the open source concept for servo applications. The most outstanding contribution of this paper is the implementation of a Genetic Algorithm for online self tuning with a focus on both high-quality servo control and reduction of vibrations during the positioning of a linear motion system. The proposed techniques have been validated on a real platform and form a novel, effective approach as compared to the conventional tuning methods that employ empirical or analytical solutions and cannot improve their parameter set. The controller was elaborated from the Graphical User Interface to the logical implementation while using free tools. This approach also allows for modification and updates to be made easily, thereby reducing the susceptibility to obsolescence. A comparison of the logical implementation with the manufacturer software was also conducted in order to test the performance of free tools in FPGAs. The Graphical User Interface developed in Python presents features, such as speed profiling, controller auto-tuning, measurement of main parameters, and monitoring of servo system vibrations.

## 1. Introduction

In the last years, the demand for motion controllers has increased in industrial applications, such as robot manipulators, computerized numerical control (CNC) machines and frequency converters [1,2]. In this context, efforts have been made by researchers to improve the performance of these control systems. However, most control systems present limitations to modify or replace control algorithms [3]. These limitations are presented by the control system architecture that can be classified as: (1) closed architecture, where it is not possible to access the control algorithm and communication protocols, (2) hybrid architecture, where the control strategies cannot be modified, but it is possible to add devices to the system, and (3) the open architecture, where both hardware and software and control strategies can be modified [2,4]. Unlike the mechanical system, the electrical and software stages become obsolete in less time, which represents an area of opportunity in servo systems [3]. The Field Programmable Gate Arrays (FPGAs) are devices with a large number of programmable logic elements, such as composite, flip-flops, and wires, and use hardware description language (HDL). Both the portability of the cores developed between families and FPGA companies and the availability of these in the market have made use of FPGAs proliferate in different areas of industry, aeronautics, robotics, and automation [5,6,7,8]. However, each manufacturer provides its software for generating the bitstream file and programming the FPGA, i.e., both the operation and distribution of logic resources are closed architectures and only large companies know how their interior works. Fortunately, a project was developed in order to synthesize and generate the bitstream file from the verilog language with free tools, with which the concept of open hardware [9] arises.

On the other hand, the Proportional-Integral-Derivative (PID) control is the most used algorithm in linear and non-linear control systems, because it provides simplicity, functionality and easy implementation [10]. In addition, a correct adjustment of the motion controller is necessary to implement speed profiling techniques that help to smooth the position, reduce sudden changes in its acceleration and avoid discontinuities that are directly related to the quality of the product developed, for example, millimeter detail of the pieces [11,12,13].

Reference [14] presented an experimental study that demonstrates the efficiency of using an FPGA in the control loop of CNC for micro-machining. The main disadvantage is the use of platforms that depend on external manufacturers and, therefore, on a specific system, such as LabVIEW. The authors of [15,16] proposed a modular control architecture with open-source electronics (for an educational manipulator robot and a CNC, respectively). One of the disadvantages is that these components are susceptible to obsolescence and require a microcontroller (MCU) for each task, such as communication, interpolation, and control loop, in addition to the dependence of the control software for the manipulator robot (Matlab software).

The authors in [5] used an FPGA to control a self-balancing robot without taking into account the analysis of robot dynamics, controller design, implementation, and tuning techniques. FPGA family ICE40 was used, whose main feature is the possibility of programming from free tools. However, the used FPGA has limitations due to its logical resources (lack of multiplying modules). References [17,18] carried out torque control experiments in stepper motors with FPGA, where both works used the Quartus software by Altera company. Nevertheless, these authors did not consider a tracking control design due to the limitations of abrupt acceleration changes.

In the works [19,20], the authors developed controllers for robot manipulators with FPGA Xilinx and digital signal processor (DSP) with the purpose of distributing tasks and not overloading the DSP. The work that was developed in [21] performs a similar application combining the use of FPGA and a MCU, where it uses the MCU as a bridge between the communication of the PC and FPGA, while the FPGA is in charge of performing the control actions. The aforementioned works lead to the increase of cost due to the use of two software owned by each digital device company. Reference [22] performs a speed control of a permanent magnet synchronous motor (PMSM) based on FPGA of the Xilinx family, where the development and simulation of cores were elaborated in Matlab. However, the work remained in simulation without realizing the implementation of the system to measure its real performance.

The development of a predictive control for direct current (DC) motors was proposed in [23], where an FPGA development board was employed from National Instruments with LabVIEW software in order to generate an industrial application. The work aims to show the ease of the graphic programming tool, as well as that it is notorious that FPGAs are an appropriate alternative for the design of control systems. It also highlights the use of electronic boards and programming software of the companies and the raised cost of the system; therefore, the use has not been explored for free tools for implementation of these systems and eliminate the dependence on design tools that become obsolete over time due to the new devices that were launched by these companies. Another feature that the aforementioned work described is the design of a methodology for the synthetization of the systems, being mostly experimental, as [24] and, in other cases, they cannot improve all of the parameters that are required for an optimal control.

The main objective of this work is to develop a self-tuning open-architecture controller that is based on open-source software and tools, thereby combating the susceptibility to obsolescence by the modularity and portability characterization of the system since. These features allow both the easy adaptation to other control applications and reduction of the retrofit costs.

The article is divided into the following sections. Section 2 describes the dynamic model of the plant and employed motion profile. Section 3 presents the controller design as well as the methodology for self-tuning servo system. Section 4 describes the implementation of the controller into a linear platform. In Section 5, the results and discussion of the experimentation are presented and, finally, in Section 6, the conclusions are given.

## 2. The Dynamic Model of a Servo System

Motors are widely used in industrial applications and control systems. Thus, it is necessary to obtain a model that allows for a correct analysis. The dynamic model of this servo system depends on the electrical and mechanical characteristics, as resistance *R*, inductance *L*, and inertia *J* of armature, the counter-electromotive force $e_{a}$, and the friction *B* [25]. Figure 1 shows the block diagram of the DC motor model with a proportional current loop. Where *K* is a proportional gain of current loop, $V_{i}$ and *I* are the voltage and current applied to the armature, τ is the torque generated in the shaft of the motor, Θ(s) and Ω(s) are the position and angular velocity of the shaft, and $K_{b}$ and $K_{m}$ are the constants of the counter-electromotive force and the torque, respectively. The electrical relationship is $\frac{1}{Ls+R}$, while and mechanical relationship is given as $\frac{1}{Js+B}$.

In the case of this servo system, the velocity has to change following a characteristic shape in order to achieve a desired position, where the movement must be less aggressive when compared to a step input to avoid a high stress on the shaft of the motor. The velocity profiles consist on different shapes, which depend on the motion; these can be triangular, trapezoidal, S-curve, or parabolic functions [26]. The triangular and parabolic velocity profiles consist of accelerating and decelerating DC motor sections without a constant velocity phase. The triangular profile is more aggressive in movement than the parabolic profile, since it changes the velocity abruptly, while the parabolic profile tends to increase and decrease the acceleration in a smoother way.

The S-curve velocity profile is a smooth trajectory compounded by a third-degree polynomial. It permits delimiting the jerk into an interval proposed by the designer in order to ensure the durability of the actuator [27]. A disadvantage presented in this kind of motion profiles is the computational performance, since it requires different techniques of derivatives and integrals [11]. On the other hand, the trapezoidal velocity profile is a second-degree polynomial that consists of an acceleration and deceleration phase; here, the velocity increase its magnitude considerably; a constant velocity stage is present in the middle of the acceleration-deceleration phases. The computational cost is suitable for simple structure and fast response [28]. That is the main reason for which the trapezoidal profile is one of the most used speed profiles in commercial controllers [11]. Its movements consist of an acceleration period $t_{0}\leq t\leq t_{0}+T_{a}$, followed by a constant speed period $t_{0}+T_{a}\leq t\leq T-T_{a}$ and finished with a deceleration period $T-T_{a}\leq t\leq T$. While the velocity profile has a trapezoidal shape, as shown in Figure 2, the evolution of position $\theta_{d}\left(t\right)$ is represented using a second-order polynomial, see Equation (1), which represents the general position.
(1)$$\theta_{d}\left(t\right)=\left\{\begin{array}{ll} \frac{a}{2}t^{2} & t_{0}\leq t\leq t_{0}+T_{a}\\ \frac{aT}{3}t-\frac{5aT^{2}}{18} & t_{0}+T_{a}\leq t\leq T-T_{a}\\ \frac{-a}{2}t^{2}+aTt-\frac{5aT^{2}}{18} & T-T_{a}\leq t\leq T \end{array}\right.$$

The maximum angular velocity $\omega_{max}$ and the desired angular position $\theta_{d}$ of the motor spur are proposed and, with this, the total duration *T* of the movement and the acceleration *a* are calculated, as follows [28]:(2)$$T=\frac{3\theta_{d}}{2\omega_{max}}$$
(3)$$a=\frac{3\omega_{max}}{T}$$

## 3. Controller and Tuning

The PID controller is the most widely used control strategy in industrial process, because of remarkable effectiveness, simplicity of implementation, and broad applicability [10,21,29,30]. The PID controller plays a vital role for the system performance by both minimizing the error and providing better systems functional stability. Below is presented the Equation of a PID controller (Equation 4) in continuous time [25].
(4)$$i_{d}\left(t\right)=k_{p}e\left(t\right)+k_{i}\int e\left(t\right)dt+k_{d}\frac{d}{dt}e\left(t\right)$$
where $k_{p}$ is the proportional gain, $k_{i}$ is the integral gain, $k_{d}$ is the derivative gain, e(t) is the error, and $i_{d}\left(t\right)$ is the control signal.

Using the backward Euler method, the discretization of the PID controller (Equation 5) was performed with a sampling time $T_{s}$. In Equation (Equation 5) e[k], e[k−1] and e[k−2] represent the current error and previous errors in discrete time; $i_{d}\left[k\right]$ and $i_{d}\left[k-1\right]$ denote the current and the delayed output of the controller [29], respectively.
(5)$$i_{d}\left[k\right]=a_{0}e\left[k\right]+a_{1}e\left[k-1\right]+a_{2}e\left[k-2\right]+i_{d}\left[k-1\right]$$
where:$$a_{0}=k_{p}+T_{s}k_{i}+\frac{k_{d}}{T_{s}},\;\;a_{1}=-\left(k_{p}+2\frac{k_{d}}{T_{s}}\right),\;\;a_{2}=\frac{k_{d}}{T_{s}}$$

The tuning process is fundamental in the design of controllers. If the model of the system is known, then it is possible to apply control design techniques such as the locus of the roots or frequency response to identify the required system specifications. However, there are also experimental techniques for tuning the PID controller; one of these is the case of Ziegler–Nichols, where the system model is not required. According to [24], the tendency on PID controllers is to solve multiple requirements, specifications, and design objectives that conflict with each other. A robust PID controller can also be solved through an optimization procedure.

A Genetic Algoritms (GA) is heuristic algorithm whose function is to evolve a set of individuals (population) and make these individuals approach the optimal solution with the passage of each generation (algorithm iteration). The main operators of GA are crossover and mutation. The individuals of the populations are improving through both the generations with these operators and selection processes. A characteristic that differentiates GA is the way of representing the problem, since it raises the modification of each individual through their chromosomes. In this case, it is represented through binary numbers. Figure 3 shows the iterative processes that are involved in GA. A mono-objective algorithm is proposed, since it requires, in addition to a zero steady-state error, that the establishment time of the system is short and that the controller response is smooth, so that the system is not saturated.

Unlike works [31,32,33], our algorithm does not run in simulation. However, it is a process that runs in line from the developed Graphical User Interface (GUI). By integrating an accelerometer in the servo system, it is also possible to measure the amplitude and corresponding frequency components of the vibrations. This allows for the incorporation of these signals as inputs to the GA in the form of a restriction. The objective function is composed of three convex cost functions: the Integral Absolute Error (IAE) Equation (Equation 6) as an indirect measure of overshoot and Total Variation (TV) Equation (Equation 7) to evaluate changes in the control signal $i_{d}\left(t\right)$. These equations are minimized by the GA algorithm. As a result, the maximum overshoot is limited and the settling and error times are minimized via a smooth control signal. Nevertheless, in order to minimize the complexity and convergence time of the algorithm, $J_{m}$ Equation (Equation 8) is described as a weighted objective function of the described equations. Moreover, α, β, and γ are the penalty parameters and their function is to control the search region by penalizing the restriction function. These values are modified through the restrictions, e.g., when a restriction leaves the search region, the objective function is strongly penalized, and, as these restrictions are met, the values of α and β remain 1, while γ remains 0.
(6)$$IAE=\int_{0}^{\infty }\left|e\left(t\right)\right|dt$$
(7)$$TV=\int_{0}^{\infty }\left|i_{d}\left(t+1\right)-i_{d}\left(t\right)\right|dt$$
(8)$$min(J_{m}=\alpha IAE+\beta TV+\gamma )$$
with restrictions:$$0<k_{p}\leq 100\;;\;0<k_{i}\leq 200\;;\;0<k_{d}\leq 10$$
$$\mu_{p}\leq 5\%$$
0.2<|nF|<1

## 4. Reconfigurable Hardware Description

The FPGAs were developed by the company Xilinx in 1984, with the purpose of creating chips that acted as a blank tape, allowing for users to program the technology themselves. The concept was successful and other companies developed FPGAs due to its reconfigurable design and parallel processing. However, as it is a new technology, each manufacturer keeps the internal performance secret, so the user has to acquire software from the manufacturer in order to perform the synthesis and program them [7]. A further drawback occurs when these companies develop a new product, so the software used is obsolete, not to mention that they need a specific processor to run these software. Until a few years ago, the IceStorm project was introduced, in which reverse engineering was carried out to obtain the bitstream of a specific family of FPGA (Lattice iCE40). This enalbed the obtainment of both the operation and internal distribution [9], thus completing the development flow from the descriptive language Verilog to the creation of the bitstream [7,8] based on free cross platform tools.

The structure of the IceStorm project is depicted with a flow chart in Figure 4. In [9], it is observed that IceStorm is an open-source project that is integrated by different tools: (1) Yosys is a synthesis tool similar to the one used by the manufacturer that converts Verilog code to a different format, such as FLIG, EDIF, BTOR, SMT-LIB, and RTL with Verilog 2005 stand. Moreover, (2) Arachne-PNR is a tool that implements the hardware build process place and route using BLIF files and generates the IceStorm TXT file. Arachne-PNR is currently unsupported and the NextPNR tool became a full functional replacement with significant enhancements. Finally, (3) IcePack converts TXT file into bitstream [9,34].

The implementation of the cores was developed in Verilog language through the IceStorm tool. An FPGA ICE40UP5K from the Lattice family was used with a clock frequency of 24 MHz. Figure 5 shows modules implemented in FPGA for both real-time control and monitoring. The UART module is responsible for communication between the FPGA and the Python GUI. Communication is initialized through the GUI in a format with the following 6-byte structure for the earnings $a_{0}$, $a_{1}$, and $a_{2}$ of the discrete PID controller, respectively, followed by a byte of actuator configuration (Digital Analog Converter or PWM module), while using 2 bytes for the number of points of the desired path and finally the points of the path to be executed. The FPGA executes the monitoring with 0.1 ms sampling time and sends a frame to the GUI every 5 ms, which contains 2 bytes for monitoring motor current, 2 bytes for X axis accelerometer data, 2 bytes for current position of the servo system, and 1 byte of the PID control signal. All of the information received by the GUI (current position, current consumed, amplitude, and vibration frequencies) are shown by means of graphs. The first 9 bytes, which correspond to the gain configuration, the PID controller output and the number of points in the path, are stored in a register bank. The next desired path bytes are stored in the RAM. The PID controller begins executing the control action when all points on the path are received. The communication is transmitted at a rate of 115,200 bits per second and 8 bits of data without parity. A module was developed for reading the encoder whose sampling is 41.6 ns and is stored every 0.1 ms in a register, the resolution of the encoder is 16 bits.

The main feature of the design is modularity, which is, if more degrees of freedom (DOF) need to be added, it is possible to clone the first DOF until the required drivers are obtained, just like the other modules. Internally, a First Input First Output (FIFO) stack module was developed to control the data flow of each of the peripherals, where each data packet is made up of a byte of control and address of the peripheral, 2 addressing bytes, and 2 bytes of information, to be able to add in the future, modules in a faster and more generic way. The MCP3208 ADC and MCP4921 DAC drivers use a separate Serial Peripheral Interface (SPI) protocol to use parallel processing. In addition, an Inter-Integrated Circuit (I2C) communication module was implemented to perform mechanical vibration monitoring with the MPU6050 sensor.

Reference [35] presents a methodology for the efficient application of control laws with one-bit signals at the input and output. The signal is directly applied to physical systems, through pulse density modulation. The main advantage is that the need for multipliers is eliminated in the control algorithm; however, the quantization levels depend on the sampling time. This methodology is an interesting option for FPGAs that not have multiplier resources. For this project, it is proposed to use the DSP resources of the FPGA ICE40UP5K. The implementation of the PID module in FPGA is obtained according to Section 3, see Equation (Equation 5). Therefore, 2 registers are used at the input of the error to make the delays of this signal. A state machine controls and synchronizes the described modules and a counter is responsible for making the appropriate changes to the multiplier input. In order to use high resolution control signal, an accumulator (register with adder) is added to the output of the DSP block. This way the output is obtained with greater sensitivity and it is not necessary to cut the length of the control signal. Thus, the number and features of the built-in multipliers or DSP segments that the device possesses is an important aspect to consider. Most FPGAs that are available in the market have multipliers with a fixed bus width, e.g., 18 × 18 bit lengths. If a custom bus width is used, then the synthesis tool will use logical resources to build a custom multiplier instead of using the available hardware-based modules. Therefore, the error signal and the controller gains must have a bus width that matches with the available built-in multipliers. In this work, multipliers of 16 × 16 bits were used. The signal error has a fixed point format of 16.0 and the gains have a fixed point format of 8.8. The *e* signal represents the error signal, which is the difference between the reference point and the feedback data. Similarly, the signals a0, a1, and a2 are the gains of PID controller, and their value depends on $k_{p}$, $k_{i}$, $k_{d}$, and $T_{s}$.

On the other hand, the Ik signal is the filter output, which is sent forward to the actuator. Moreover, CLK, RST, ST_PID, and EO_PID denote the clock signals and master reset, process start, and process end, respectively. It should be noted that free tools do not perform an optimal inference of the DSP blocks contained in the FPGA ICE40UP5K. For this reason, an internal DSP module is used to make the PID structure as depicted in Figure 6 and is configured to perform signed multiplications and accumulate at the output of the multipliers; these sums are unsigned. Figure 6 presents the C1 and C2 parameters that are used to pass the controller error and gains through a parallel register, and C7 enables the use of a 32-bit register for multiplier output, taking them to high level. Bits C11, C10, and C18, C17 enable the high part (16 most significant bits) and low part (16 least significant bits) to be the input of the internal adder and they are configured taking C11 and C18 high, and C10 and C17 low. Bits C12 and C19 are kept low in order to configure the DSP block as accumulator. Bits C9, C8, and C16, C15 provide the output as a registry output that has the accumulator function. These are configured by taking the C9, C16 bits low and the C8 and C15 bits high. Bits C24 and C23 are raised high to configure a signed multiplier. In addition to the configuration of the DSP block, the state machine is a fundamental part for the correct operation of the PID controller. One of the advantages of free tools is the possibility of configuring in ways that the tools of manufacturer do not allow.

## 5. Results and Discussion

In order to observe the performance of the open source synthesizer for the FPGA, Table 1 shows a comparison with the logic resources used by the FPGA in similar controllers and the proposed open source tool (IceStorm project) was made when implementing the design of the controller in the Lattice family of the ICE40UP5k model. This comparison presents that the open source tool is a good alternative for reconfigurable digital design in control applications, such as servo systems.

It is necessary to perform a simulation in GTKWave that allows for the debugging of the design when processing in hardware is evaluated. Figure 7 presents the simulation results of a good performance in reconfigurable digital design. In this simulation, the position control of a servo system is shown, the reference signal is observed (refp[15:0]), the position followed by the system (theta[15:0]), the error calculated by the system (ek[15:0]), and the controller output (ik[23:0]). This simulation demonstrates that the processing units or blocks function correctly and that the formats of the fixed point units were selected properly.

Table 1 shows the logical resources of the total system implementation. The last column also highlights the minimum periods (maximum frequency) of operation of the controller. The proposed system has a main clock source of 24 MHz and the sampling time is 0.1 ms. The time analyzer of the synthesis tool (synthesized hardware and hardware routing) showed that the most critical path has a latency of 32 ns; therefore, the system executes the control algorithms without difficulties.

The industrial linear actuator that was used in this research work was disposed from a factory and acquired by Universidad Autónoma de Querétaro as a second-hand industrial system, and no control unit was included with the purchase. Therefore, the aim of this project was to design a new control system based-on open source modern technology. Within the project three primary requirements were defined: low cost, open architecture, and the path tracking error should be lower than 2% velocity of 200 $\frac{\mathrm{mm}}{s}$. The proposed architecture is comprised of a standard PC or micro-PC and an FPGA-based development board. The FPGA is responsible for decoding feedback position data, computing the control output and commutation signals, and transmitting data to the actuator, whereas the PC interpolates movements, and transmits the position reference, while it receives its current and position. In Table 2, the parameters of both the brushless motor and analog servo drive are shown. The servo driver requires one analog current reference. Current reference is given as voltage. A set point of +10 V represents a current of 6 A at a given motor coil. An analog converter MCP4921 is used to generate such current references. However, the DAC output ranges from 0 to 5 V. Therefore, the PCB include a set of operational amplifiers to rearrange the output from −10 V to +10 V.

The analysis of vibrations allows for identifying physical problems in the components of rotating machines. These issues can be detected by means of a series of non-destructive techniques of data collection. Vibration sensors, such as the accelerometer, are suitable for different industrial applications [36]. When the vibration signal is monitored, it is presented in the form of simple harmonic movement that is, in terms of variation, in the amplitude of the vibration signal [37]. To interpret the signal, different processing techniques have been used, such as the Fast Fourier Transformation (FFT) and the spectrogram algorithms [38]. With these techniques, it is possible to observe the state of a rotary machine, i.e., the failure of cracks and seizures can be identified based on the main frequency range of the machine engine [36,39,40,41].

The GUI developed in Python executes the calculations for the trapezoidal velocity profile. This interface is responsible for modifying the discrete PID controller gains and sending these values $\theta_{d}$, $a_{0}$, $a_{1}$, and $a_{2}$ via the serial port to the FPGA. The GA is executed in the GUI as well. In addition, the GUI is responsible for receiving the information from the FPGA, the values of the X axis of the sensor, the current position of the motor, the error, and the control signal. These variables are plotted to observe the behavior and performance of both the controller and translational mechatronics system. The data can be stored by date at each start-up of the system. Figure 8 shows the proposed control and self-tuning scheme. The Algorithm 1 presents the steps for the implementation of the GA.

**Algorithm 1:** Pseudocode of the GA.

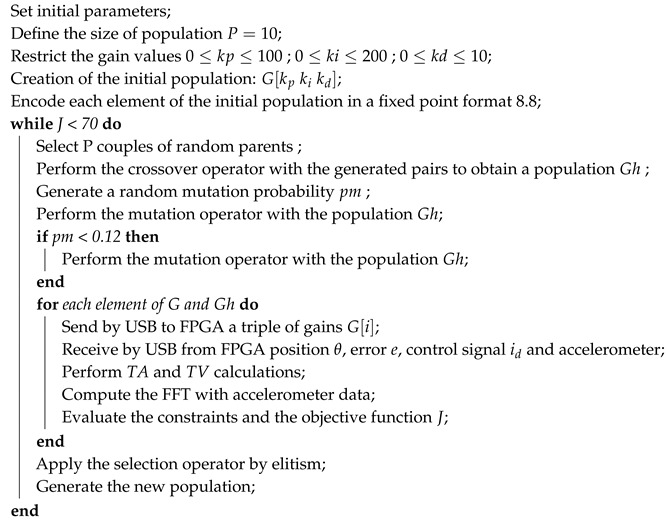



Figure 9a shows the response of the servo system θ, where the control signal is $i_{c}$, while the reference that follows the system is denoted with $\theta_{d}$ and its value is 5π radians, which is equivalent to 50 mm of advance applied to the GA for autotune. Figure 9b shows the response of the objective function through the different generations. According to the configuration proposed, the number of generations is 50; however, it is observed that from generation 20 a minimum value constant of the objective function is reached and the response of the servo system is as desired according to Figure 9a. This means that the servo system establishment time is smaller than 200 ms, the steady state error is zero, and the maximum overshoot is 1.6%. In addition, the response of the control signal is smooth and not saturated.

During the experiment, 15 replicates were carried out, consisting of the estimation of the parameters $a_{0}$, $a_{1}$, and $a_{2}$ of the discrete PID controller using the proposed GA. The replicates record a certain number of iterations before the algorithm stopped due to the established stop condition. The values reached the objective function, and the restrictions posed, as well as an indicator of whether a desirable, feasible, or no solution was found. It is desired to achieve the setting time lower than 200 ms and with less than 5% overshoot. First, a statistical analysis of the completion of the algorithm was performed. Figure 10a depicts the distribution of the number of iterations that are required before the algorithm stopped by the magnitude of the objective function less than 70, according to the literature. Because there is a bias to the right, the median is a useful statistic tool for describing the centrality of the observations, which can be interpreted as the number of iterations required for the algorithm convergence.

In this sense, the median of the necessary iterations was found to be 26 iterations, and using Bootstrap the normal asymptotic confidence interval was estimated at 95% of this statistic; it turned out that the average number of iterations required for convergence was between 19 and 33 iterations (the exact interval was 19.29 to 32.71). Another interesting remark is that, even if the algorithm stops due to an established stop condition, this does not guarantee at any time that a feasible or desirable result will be found. In addition, it was found that the algorithm did not find a viable solution 3.22% of the time, 44.84% a feasible but unwanted solution was found, and 54.84% of the time, a possible and desired solution was reached (Figure 10b). According to the replicas, the 95% confidence intervals were calculated for the event where an at least feasible solution was found and another for the event, where an achievable and desired solution is found. In this way, the algorithm obtains a 95% confidence and the desired solution between 36.30% and 72.22% of the time, and at least a feasible solution between 81.49% and 99.83% of the time.

Regarding the objective functions, Figure 10c shows that the set of feasible and desirable solutions achieves highly variable objectives, since the estimated standard deviation of the aim achieved in this type of solution is 9.3468 units. While the other types of solutions have smaller variations, furthermore, Figure 10c shows that the cube of solutions that limits the set of feasible and desirable solutions contains in its way only possible solutions and not viable solutions. Additionally, Figure 10d highlights the values that were reached by the objective function in the first generation of the algorithm and the last age of the algorithm. It can be observed that the algorithm’s solutions are more variable at the end of its execution and that the range of the initial objective is contained in the final goal, which may be an explanation of the rapid convergence of the algorithm for this particular case. In this sense, Figure 10e displays the change between the value of the objective function at the beginning of the algorithm and at the end of it, where it can be shown that most of the changes are concentrated around 0. Thus, when applying a paired t-test to contrast the hypothesis that statistically, the algorithm does not generate the value of the objective function with the assumption that if there is a statistically significant change, it was found that the mean estimated change was −1.8288 units; but there is not enough statistical evidence to rule out the hypothesis that statistically the algorithm does not generate the value of the objective function (where the calculated t-statistic was −0.6733 with 14 degrees of freedom and a *p*-value of 0.5117). However, this was not repeated in both restrictions.

Figure 10f shows that the ranges of the IAE restriction reached by each objective are mutually exclusive. At the same time, Figure 10g depicts that the behavior observed by the restriction TV was similar to that observed in the objective function, where it is also observed that the cube of solutions that limits the set of feasible and desirable solutions contains in its way the only possible solutions and not viable solutions. Nonetheless, as in the case of the objective function, it was not found that there was a statistically significant change in the restrictions before and after the algorithm. In the case of the IAE restriction, a mean shift in −1.9892 was estimated with an associated t statistic of −0.6256 of 14 degrees of freedom with a *p*-value of 0.5417; and, for the case of the TV restriction, a mean change of 0.1603393 was estimated with an associated t statistic of 0.0618 of 14 degrees of freedom and a *p*-value of 0.9516.

The algorithm did not find a feasible solution 3.22% of the time, as shown in the aforementioned statistical analysis. In case the algorithm is not converging, the outcome is visually manifested, since both large energy consumption and vibrations present in the system. Therefore, it is possible to implement an emergency stop function in the GUI for safety as in any control system.

The validation with the simulation was conducted once the operation of the controller was implemented in the FPGA. Two tests were performed to observe the behavior of the sensor when steps with a certain magnitude are supplied as inputs without GA. Finally, the behavior is compared to the case when the input is supplied via trapezoidal speed profiles with GA. Figure 11a shows the response of the system; the reference values applied in the control were 25π, −25π, and 0 rad equivalent to 250 mm, −250 mm, and 0 mm. It is observed that the system is able to reach the reference values; however, it suffers from very abrupt changes, which can be seen in Figure 11c. Figure 11b shows the desired speed and angular velocity that were generated by the system. Moreover, Figure 11d highlights the system error when both step inputs and control signal were supplied to the system.

In the following test, the trapezoidal velocity profile of three points (25π, −25π and 0 rad equivalent to 250 mm, −250 mm and 0 mm) with a maximum speed of 20π
$\frac{\mathrm{rad}}{s}$ were applied. Figure 12a,b show the system behavior when the reference in position and speed are applied, respectively. It is also observed in Figure 12a that the magnitude of the error decreased considerably when compared to the previous test. The control signal is kept at desired low levels, taking into account that the maximum current is 4.9 A. Figure 12d provides important information regarding the smooth movements generated by the proposed velocity profile.

Figure 13a,b show the frequency analysis via the FFT algorithm output when step inputs and trapezoidal velocity profile were applied, respectively. When the system presents step inputs, the vibrations of the system increase, which may cause failures in the future.

When a speed profile is applied these vibrations tend to decrease considerably. This way, greater performance is supported and a longer life time is provided for the translational mechatronics system. According to the tests, the first case employed conventional tuning method. However, only trajectory tracking of servo systems is not sufficient for control purposes (i.e., reaching the reference in a short time), since, if large displacements are desired, then the settling times are affected between 1 and 3 s, moreover, the overshoot becomes 75% and even 160% during direction changes (see Figure 11a). Figure 11d highlight that increased speeds and accelerations generate jerks, so the energy consumption increases significantly and sometimes demands 100% power supply (saturation 4.9 A and −4.9 A). Nevertheless, if the proposed auto-tuning algorithm is implemented and a movement profile is applied to limit the maximum speeds of the system, then the maximum error of reference tracking is 0.15 rad or 0.17%, as shown in Figure 12a. In this case, the system speed is limited to the defined maximum speed $\omega_{max}=20\pi \;\frac{\mathrm{rad}}{s}$ and follows the velocity profile reference with an overshoot less than 2%, as shown in Figure 12b. This reduces the maximum current draw by 92% (0.4 A and −0.4 A), as shown in Figure 12d. Based on the response of the accelerometer, a comparison between the tests were carried out in Figure 13. The frequencies that were greater than 60 Hz were eliminated by applying the tuning algorithm and movement profiles. The 55 Hz frequency component of 0.4 magnitude found in the first spectrum was also eliminated. Moreover, the low frequency components (i.e., between 1 and 60 Hz) of magnitude less than 0.2 were also significantly attenuated in the spectrum.

## 6. Conclusions

This paper presents the development and update of a control system employed for a translational mechatronics apparatus (servo system) based-on open source software and hardware tools. The work shows the complete development and analysis of the servo system model, from the validation of the correct operation of the controller through simulation, to solutions for the implementation of the designed controller. Furthermore, a methodology was developed for the auto tuning of the employed PID controller for efficient trajectory tracking in the analyzed linear movement system. The control system was implemented in a low-cost FPGA ICE40UP5k of the Lattice family. All of the proposed modules are portable for any FPGA manufacturer. According to Table 3, a count was made of the control applications where an update of the control system was performed, showing that this is one of the first works where open source tools were used for programming FPGA. In addition, such methodology was elaborated, which performs on-line auto-tuning of the servo system with vibration monitoring.

According to Table 1, it is also observed that open source tools have a higher performance, because of distribution and management of logical resources. Additionally, it has the advantage of being multiplatform, that could be executed on Linux, Windows or MAC OS operating systems. A free code GUI was developed in Python for the monitoring of both system variables and vibrations generated by the mechanical system. This GUI is also responsible for both configuring the gains through serial communication and calculating the trapezoidal velocity profiles. The profile calculation algorithm can be easily replaced to test different velocity profiles. The cost of control board is around $25 dollars. Figure 12 shows that the PID controller is able to follow any path for the translational mechatronics system with an error less than 0.2%. The vibration monitoring system offers an alternative for detecting faults in the operation of the translational mechatronics system. This feature can be used as a type of predictive maintenance. As future work, we are working on implementing two more axes to complement the control of the coordinate table and implement GA on FPGA for different lineal control techniques.

## Figures and Tables

**Figure 1 sensors-20-06155-f001:**
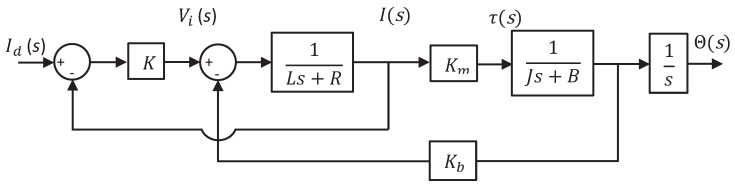
Block diagram of the simplified direct current (DC) motor model in a proportional current control scheme.

**Figure 2 sensors-20-06155-f002:**
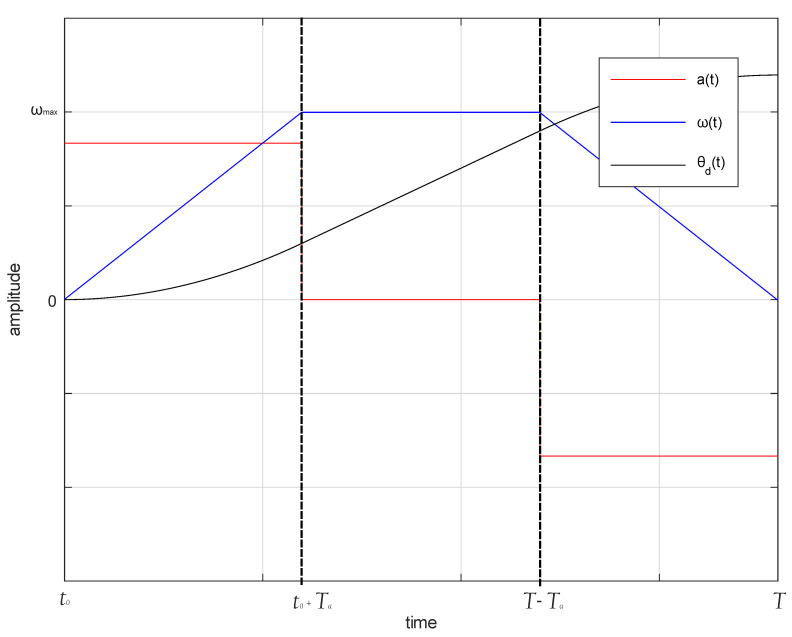
Trapezoidal velocity profile.

**Figure 3 sensors-20-06155-f003:**
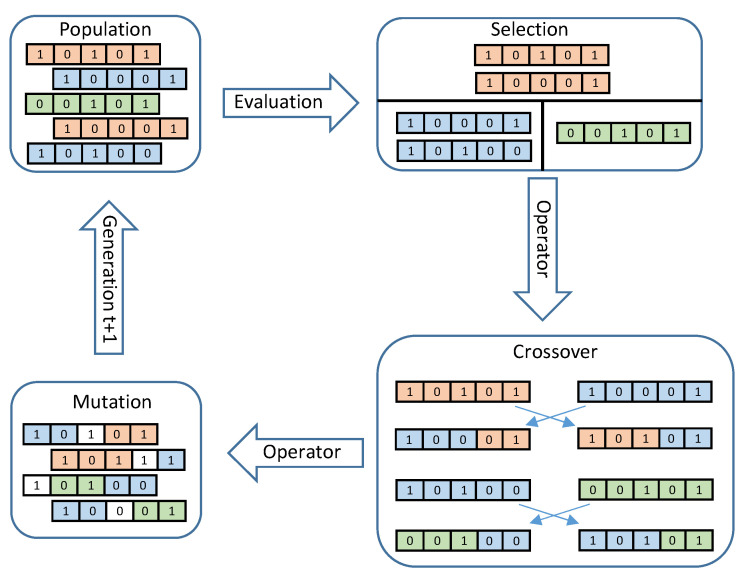
Flow chart for Genetic Algoritms (GA) process.

**Figure 4 sensors-20-06155-f004:**
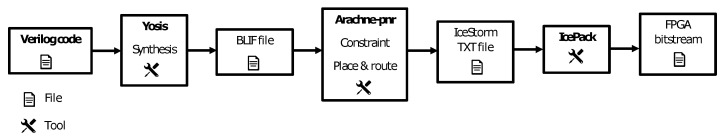
IceStorm design flow.

**Figure 5 sensors-20-06155-f005:**
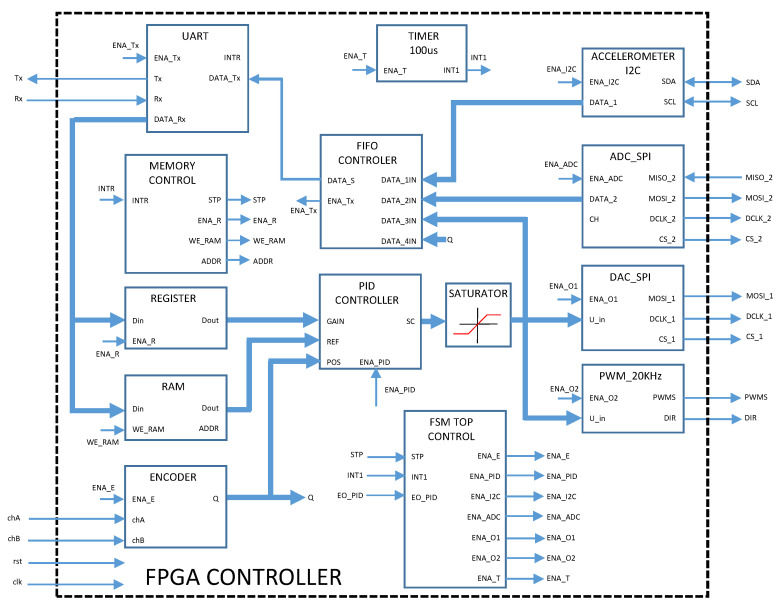
Modules implemented in the Field Programmable Gate Arrays (FPGA).

**Figure 6 sensors-20-06155-f006:**
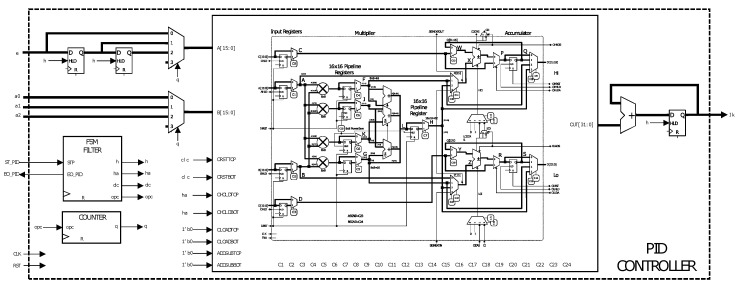
Proportional-Integral-Derivative (PID) controller implemented in the FPGA.

**Figure 7 sensors-20-06155-f007:**
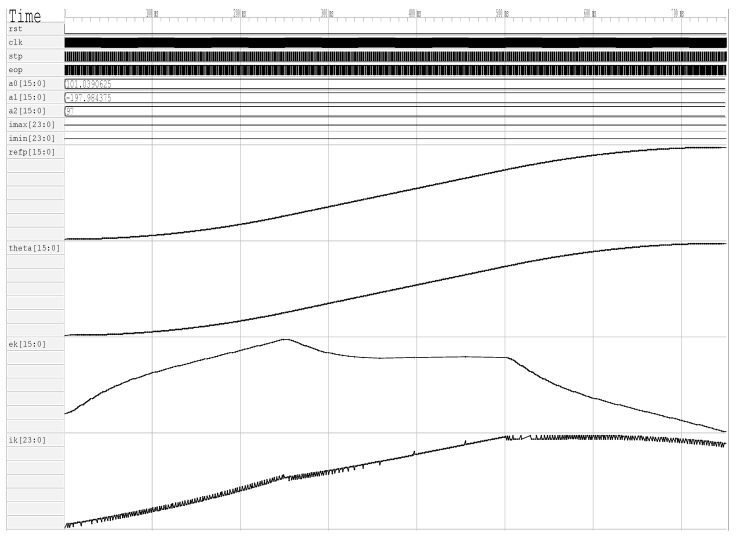
PID controllers simulation with open source tool.

**Figure 8 sensors-20-06155-f008:**
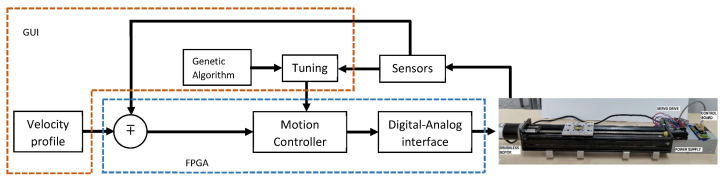
Methodology for control and self-tuning.

**Figure 9 sensors-20-06155-f009:**
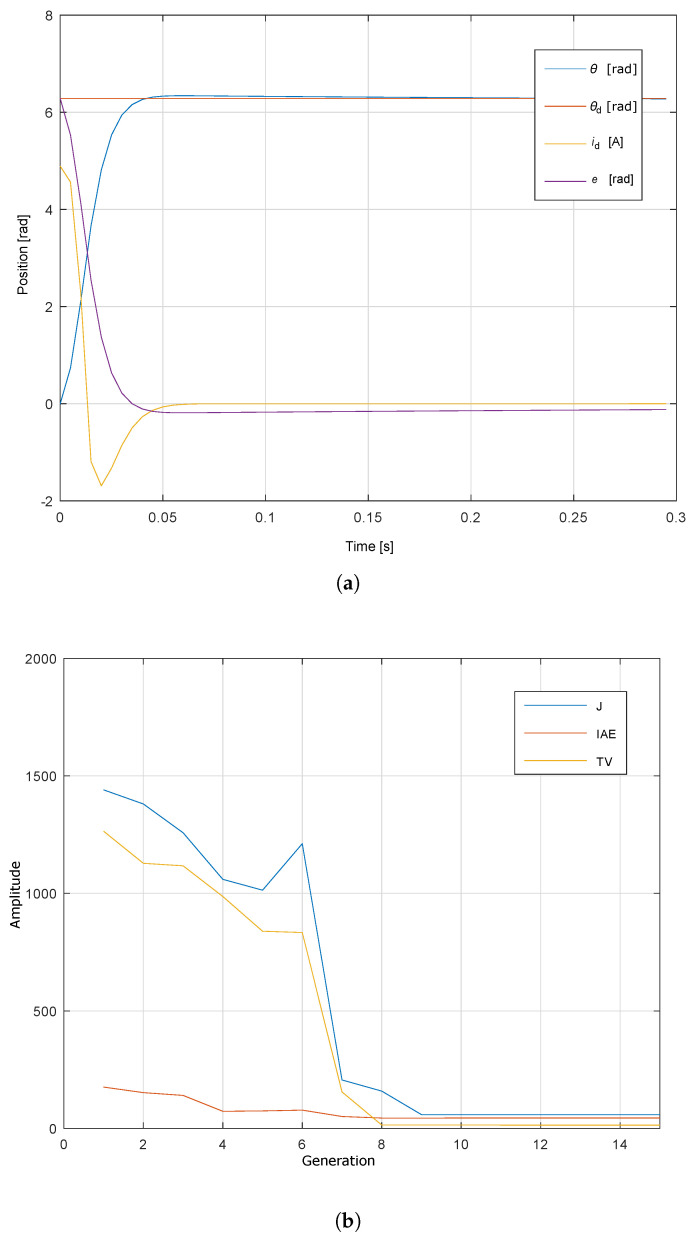
Response of GA self-tuning PID controller. (**a**) Angular position, current and error of test bench. (**b**) Objective function through generations.

**Figure 10 sensors-20-06155-f010:**
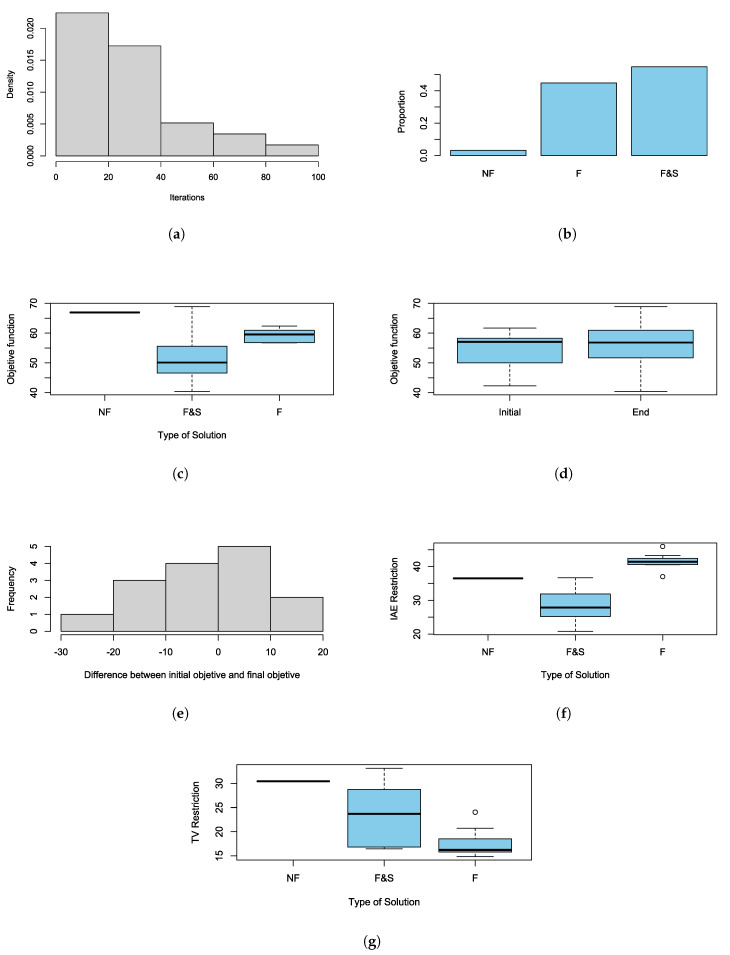
Statistical analysis of the proposed GA (**a**) distribution of the number iterations, (**b**) analysis of not feasible (NF), feasible (F) and feasible and suitable solution (F&S), (**c**) estimated standard deviation of the objective reached, (**d**) obtained values of the objective function, (**e**) change of the objective function, (**f**) change of the Integral Absolute Error (IAE) restriction, and (**g**) change of the TV restriction.

**Figure 11 sensors-20-06155-f011:**
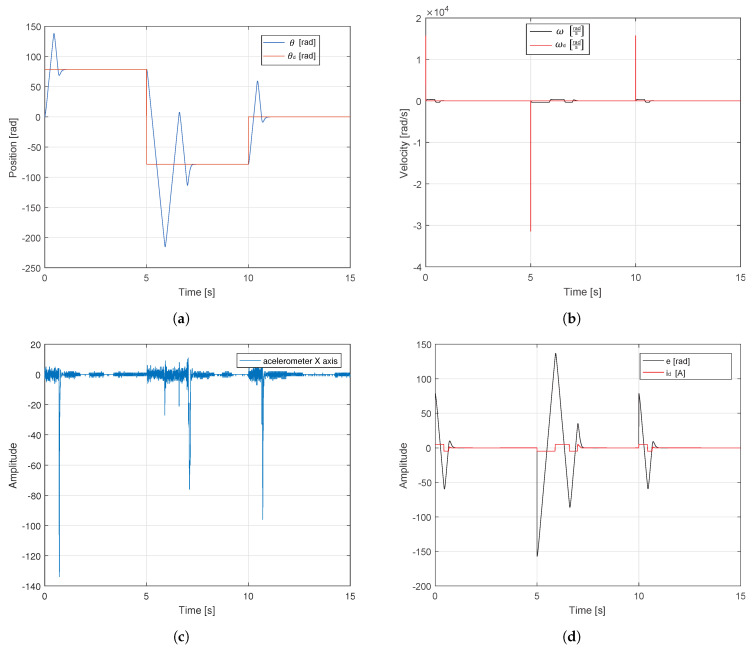
Response of test bench with a step input applied without GA applied. (**a**) Angular position and reference. (**b**) Angular and desired velocity. (**c**) Accelerometer X axis. (**d**) Error and control signal.

**Figure 12 sensors-20-06155-f012:**
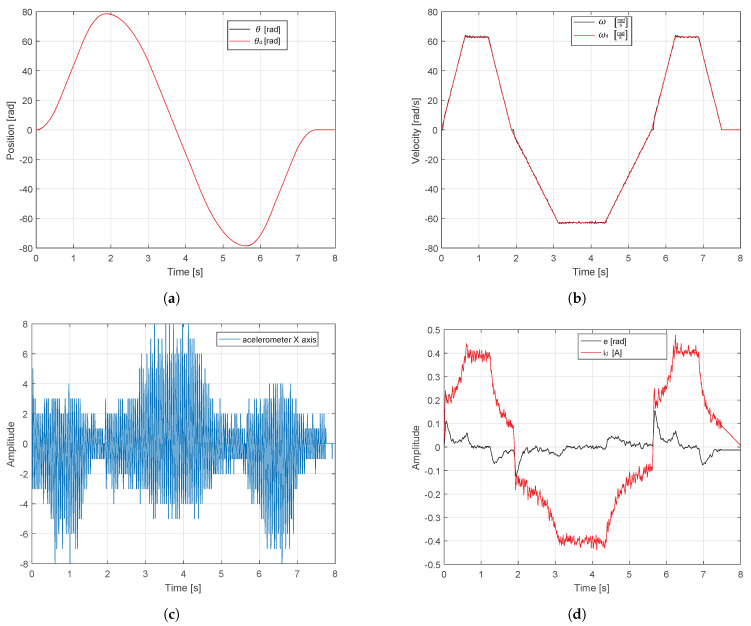
Response of test bench with a trapezoidal velocity profile and GA applied. (**a**) Angular position and reference. (**b**) Angular and desired velocity. (**c**) Accelerometer X axis. (**d**) Error and control signal.

**Figure 13 sensors-20-06155-f013:**
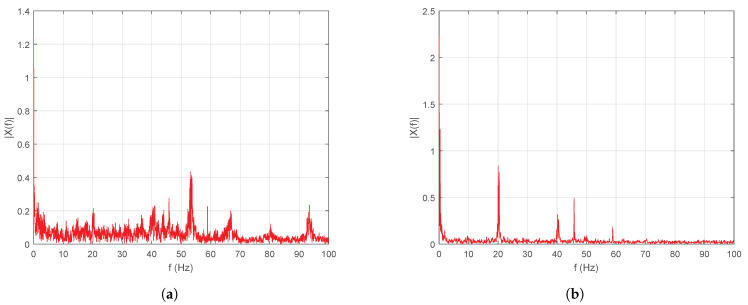
Response of accelerometer with the Fast Fourier Transformation (FFT) applied. (**a**) Input step without the GA. (**b**) Trapezoidal velocity profile applying the GA.

**Table 1 sensors-20-06155-t001:** Logic resources used.

Work	Manufacturer	Device	Flipflops	LUTs	Multipliers/DSP	SPRAM256K	HFOsc	Minimum Period
Proposed	Lattice	ICE40UP5k	963/5280	821/5280	1/8	2/4	1	37.2 ns
[1]	Xilinx	Spartan-3	2998/15,360	3792/15,360	24/24	-	-	12.174 ns
[2]	Xilinx	Spartan-3	-	-	-	-	-	-
[3]	Altera	Cyclone II DE2	1788	998	35/70	-	-	-
[5]	Lattice	ICE40hx4k	-	-	-	-	-	-
[11]	Xilinx	Zynq XC7Z01	-	-	-	-	-	-
[14]	Xilinx	Spartan-3	-	-	-	-	-	-
[17]	Altera	Max10	3079	5017	6	-	-	-
[26]	Xilinx	Zynq 7	-	-	-	-	-	-

**Table 2 sensors-20-06155-t002:** Parameters of test bench.

BLM-N23-50-1000-B Brushless Motor			B12A6 Advanced Motion Control Analog Servo Drive		
Parameter	Value	Unit	Parameter	Value	Unit
Torque Constant	0.08	$\frac{\mathrm{Nm}}{A}$	DC Supply Voltage Range	20–60	V
Continuous Torque	0.39	Nm	Maximum Peak Output Current	12	A
Peak Torque	0.83	Nm	Maximum Continuous Output Current	6	A
Continuous Current	4.9	A	Maximum Continuous Output Power	342	W
Peak Current	10.4	A	Maximum Power Dissipation	18	W
Moment of Inertia	2.5 ×$10^{-5}$	${\mathrm{kgm}}^{2}$	Switching Frequency	33	kHz
Recommended Supply Voltage	48	V	Command Sources	±10	V
Maximum Speed	523.6	$\frac{\mathrm{rad}}{s}$	Modes of Operation	current	A
Encoder Resolution	1024	PPR			

**Table 3 sensors-20-06155-t003:** Background of open controllers.

Work	Controller	Manufacturer	Programming	Tuning	Sample Time
Proposed	FPGA ICE40 UltraPlus 5k	Lattice	Open tool	GA	0.1 ms
[1]	FPGA Spartan-3	Xilinx	ISE-Xilinx	Empiric	1 ms
[2]	FPGA Spartan-3	Xilinx	ISE-Xilinx	-	-
[3]	FPGA	Altera	MATLAB/Simulink	-	-
[5]	FPGA ICE40hx4k	Lattice	Open tool	Empiric	-
[11]	Raspberry Pi-FPGA ZYNQ 7	Xilinx	Vivado-Xilinx	-	-
[14]	FPGA Spartan-3	Xilinx	LabVIEW NI	Empiric	0.1 ms
[15]	PC-Arduino	Arduino	MATLAB/Simulink	pole assignment	60 ms
[16]	PC-Launchpad-Arduino	Texas Instruments	-	-	50 ms
[17]	FPGA Max10	Altera	NiosIIsoft	-	0.2 ms
[26]	Raspberry Pi-FPGA ZYNQ 7	Xilinx	Vivado-Xilinx	Fuzzy	5 ms

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
