# Peer review of "A New Methodology for a Retrofitted Self-tuned Controller with Open-Source FPGA"

_sensors, 2020, doi:10.3390/s20216155_

Round 1

Reviewer 1 Report

In general, I think that the topics covered in this work are valid and consistent with the issues of the journal. In my opinion, it is well written, structured clearly and sequentially, and it is relatively intuitive and understandable.

However, to improve the quality of this work, I suggest that authors carefully review the text (according to the journal template) by checking the grammar and correcting the few errors and inaccuracies found in the document (for instance, the punctuation error ",." in line 56 of page 2, or the title of chapter 4, in line 161 of page 7, which is written partly in Spanish).

In particular, I think it is mandatory to evaluate the following issues:

  1. The authors should carefully review the whole paper strictly respecting the journal template (text layout, paragraph spacing, figure captions, correct placement of figures and equations in the text, style of the References section, etc.);
  2. Regarding the dynamic model of a servo system considered, the authors should provide more information to better frame the type of device used and its numerical implementation. In fact, to effectively model this system (and evaluate its behavior), I think that it is necessary to specify the type of machine adopted (e.g. type of electrical motor) and the simplifying hypotheses that have been introduced to define its digital twin (e.g. assumptions, simplifications, nonlinearities, etc.);
  3. As for the development and fine-tuning of detailed high-fidelity numerical models, capable of representing reliable digital twins of the considered physical systems, I would suggest that the authors integrate their work with some other specific bibliographic references. For example, models of this type have been presented in:
    C. Berri, M. D. L. Dalla Vedova, P. Maggiore – A Lumped Parameter High Fidelity EMA Model for Model-Based Prognostics – Proc. of the 29th European Safety and Reliability Conference (ESREL 2019), Hannover, Germany, September 22-26, 2019, No. 0480, pp. 1086-1093, ISBN: 978-981-11-2724-3
  4. Figure 7 has a very low resolution and poor readability: authors should replace it with a higher resolution schematic and, if possible, adopt larger fonts;
  5. The graphs shown in Figure 8 are too small and, especially in the paper version of the document, are poorly legible: authors should enlarge figures, legend, labels, and fonts;
  6. Note also that the label of the first graph of Figure 8, namely figure 8(a), is partially cut off and therefore cannot be read correctly;
  7. The graphs shown in Figures 9 and 10 are too small and, especially in the paper version of the document, are poorly legible: authors should enlarge figures, legends, labels, and fonts
  8. Given that several acronyms are introduced in this work, I think that a summary table explaining all these symbols/acronyms/abbreviations would be beneficial for improving its readability and comprehensibility. Therefore, I suggest including this list of symbols/acronyms in a dedicated section of this paper.

Author Response

REVIEWER 1

In general, I think that the topics covered in this work are valid and consistent with the issues of the journal. In my opinion, it is well written, structured clearly and sequentially, and it is relatively intuitive and understandable.

However, to improve the quality of this work, I suggest that authors carefully review the text (according to the journal template) by checking the grammar and correcting the few errors and inaccuracies found in the document (for instance, the punctuation error ",." in line 56 of page 2, or the title of chapter 4, in line 161 of page 7, which is written partly in Spanish).

Thank you very much for your feedback. All the comments have been attended. 

In particular, I think it is mandatory to evaluate the following issues:

  • The authors should carefully review the whole paper strictly respecting the journal template (text layout, paragraph spacing, figure captions, correct placement of figures and equations in the text, style of the References section, etc.);

Thank you very much for your observation. We have carefully reviewed the whole paper strictly respecting the journal template.

  • Regarding the dynamic model of a servo system considered, the authors should provide more information to better frame the type of device used and its numerical implementation. In fact, to effectively model this system (and evaluate its behavior), I think that it is necessary to specify the type of machine adopted (e.g. type of electrical motor) and the simplifying hypotheses that have been introduced to define its digital twin (e.g. assumptions, simplifications, nonlinearities, etc.);

Thank you very much for your commentary. Regarding the dynamic model of a servo system considered, we’ve provided additional information to better frame the type of device used, moreover numerical implementation has been described in detail. This can be read in lines 89-98.

  1. As for the development and fine-tuning of detailed high-fidelity numerical models, capable of representing reliable digital twins of the considered physical systems, I would suggest that the authors integrate their work with some other specific bibliographic references. For example, models of this type have been presented in:
    C. Berri, M. D. L. Dalla Vedova, P. Maggiore – A Lumped Parameter High Fidelity EMA Model for Model-Based Prognostics – Proc. of the 29th European Safety and Reliability Conference (ESREL 2019), Hannover, Germany, September 22-26, 2019, No. 0480, pp. 1086-1093, ISBN: 978-981-11-2724-3

We appreciate the comment. We’ve extended our work with some other specific bibliographic references as well.

  • Figure 7 has a very low resolution and poor readability: authors should replace it with a higher resolution schematic and, if possible, adopt larger fonts;

Thank you very much for your comment. Figure 7 has been updated to high resolution and best readability.

  • The graphs shown in Figure 8 are too small and, especially in the paper version of the document, are poorly legible: authors should enlarge figures, legend, labels, and fonts;

Thank you very much for  your comment. Figure 8 has been regenerated to high resolution best readability.

  • Note also that the label of the first graph of Figure 8, namely figure 8(a), is partially cut off and therefore cannot be read correctly; 

Thank you very much for your comment. Figure 8a has been updated to best readability. The plots are vectorized in order to check all the details.

  • The graphs shown in Figures 9 and 10 are too small and, especially in the paper version of the document, are poorly legible: authors should enlarge figures, legends, labels, and fonts.

Thank you very much for your comment. Both Fig. 9 and 10 have been updated to high resolution and best readability. The plots are vectorized in order to check all the details.

  • Given that several acronyms are introduced in this work, I think that a summary table explaining all these symbols/acronyms/abbreviations would be beneficial for improving its readability and comprehensibility. Therefore, I suggest including this list of symbols/acronyms in a dedicated section of this paper.

We appreciate the comment. We have included the list of acronyms and symbols.

Reviewer 2 Report

The article deals with a very interesting case-study on control application, by using the GA for tuning the PID control parameters on FPGA.

As authors state, the novelty is to employ multi-objective optimization by 3 convex cost functions. I disagree with this statement, since such approach is far from being a novelty, and indeed, if all three cost function are composed into a single, then this is not a multi-objective optimization.

In the Experiments section, the algorithm setup with appropriate parameters outlined is missing. Also, the pseudocode of the GA (many times denoted by AG, please unify this) should be written in algorithmic mode. The experimental setup should be, instead of giving a dedicated a whole section (Instrumentation”), given a dedicated table. I cannot find the upper limits (upper bounds) of the control parameters (kp, ki, kd) that are modified using the GA. Additionally, experimental section could be written more structurally and figures should be equipped with Latex signs of Greek words, of better quality, greater fonts of labels, grid exposed,... Please have a look at an example of such organisation:

https://www.sciencedirect.com/science/article/pii/S1568494620304981

I still have many questions: I wonder, if it is possible to derive any statistical analysis? How about to visualize the trend of falling value of cost functions? What are the “target applications” of your approach? What about the robustness of your approach? What happens if GA traps into a local optima, or diverges? How long does it take to re-adjust to changes in control plants? How about completely new plant? How is the initialization of PID controller performed? What was the criteria of selecting GA?

Avoid defining short-forms in Abstract, e.g. GA, GUI,... Please verify that Fig. 9 is based on the real hardware, not simulations. Why is the resolution of lines so low? Control responses of test bench without GA (Fig. 9) are very poor. Have you recorded the responses on step function with GA applied? These are in effect most intensive when testing controllers.

Please correct many Spanish words, such as “Descripcion del Hardware Reconfigurable”, or “Evaluates de population”,….

Author Response

REVIEWER 2

The article deals with a very interesting case-study on control application, by using the GA for tuning the PID control parameters on FPGA.

As authors state, the novelty is to employ multi-objective optimization by 3 convex cost functions. I disagree with this statement, since such approach is far from being a novelty, and indeed, if all three cost function are composed into a single, then this is not a multi-objective optimization. 

Thank you very much for your comment. The changes were done with a mono-objective approach.

In the Experiments section, the algorithm setup with appropriate parameters outlined is missing. Also, the pseudocode of the GA (many times denoted by AG, please unify this) should be written in algorithmic mode. The experimental setup should be, instead of giving a dedicated a whole section (Instrumentation”), given a dedicated table. I cannot find the upper limits (upper bounds) of the control parameters (kp, ki, kd) that are modified using the GA. Additionally, experimental section could be written more structurally and figures should be equipped with Latex signs of Greek words, of better quality, greater fonts of labels, grid exposed,... Please have a look at an example of such organisation:

https://www.sciencedirect.com/science/article/pii/S1568494620304981

Multiple changes have been made. Namely, we’ve eliminated the instrumentation section and added a table with the characteristics of the test bench (see Table 2). Moreover, we both modified the pseudocode (in section 4) and added the initial configuration of the algorithm parameters (see the lines of algorithm 1). The Results section and the corresponding figures (Figures 9, 11, and 12) were modified as well.

I still have many questions: I wonder, if it is possible to derive any statistical analysis? How about to visualize the trend of falling value of cost functions? What are the “target applications” of your approach? What about the robustness of your approach? What happens if GA traps into a local optima, or diverges? How long does it take to re-adjust to changes in control plants? How about completely new plant? How is the initialization of PID controller performed? What was the criteria of selecting GA?

 In the Results section, we’ve added the required statistical analysis. Moreover, we’ve both modified the pseudocode (see in section 4) and added the initial configuration of the algorithm parameters (see the lines of algorithm 1).

Avoid defining short-forms in Abstract, e.g. GA, GUI,... Please verify that Fig. 9 is based on the real hardware, not simulations. Why is the resolution of lines so low? Control responses of test bench without GA (Fig. 9) are very poor. Have you recorded the responses on step function with GA applied? These are in effect most intensive when testing controllers.

The aforementioned suggestions have been incorporated in the manuscript. Fig. 9 is not a simulation, instead, it depicts the response of the real test bench. The resolution can be seen in this way due to the sampling time with which the data is plotted (5 ms) and the magnitude of the input step function that is relatively large (250 mm) compared to the advance of the system (20 mm per revolution). In servo systems speed control is realized in order to maintain the good condition of the actuators. It has been sought to contrast the results of the accelerometer, thereby having an acceleration control via the speed profiles. 

Please correct many Spanish words, such as “Descripcion del Hardware Reconfigurable”, or “Evaluates de population”,….

Thank you for your observation, the suggested changes have been made. 

Reviewer 3 Report

First, this paper requires extensive English editing. The current version contains many typos and grammar errors.

Second, the innovation of this work is not clear. The control algorithms, FPGA implementation are quite common nowadays. The interesting part may be the so-called open architecture controller using open source software. However, this part is not discussed in this paper.

Third, it seems this work is constrained by a specified FPGA device, which is the Lattice ICE40UP5K. At the control algorithm introduced in this paper is based on GA and PID. As the authors propose an open architecture, can the users choose different FPGA devices and control algorithms?

Fourth, how the bitstream is generated? It is not clear how the control algorithm/software is converted to hardware description language and sythesized. It may be more interesting to introduce the open source synthesizer if it is developed by the authors.

I won't comment on the experimental results as they do not support the authors' original idea. At the same time, the presented architecture is quite limited targeting a specified step motor control, which a not generic.

Author Response

REVIEWER 3

First, this paper requires extensive English editing. The current version contains many typos and grammar errors.

Thank you very much for  your comment. A native English speaker revised the manuscript.

Second, the innovation of this work is not clear. The control algorithms, FPGA implementation are quite common nowadays. The interesting part may be the so-called open architecture controller using open source software. However, this part is not discussed in this paper.

In the section Reconfigurable logic description we have added a description of the bit stream. You can check the innovation of the work in lines 80-83.

Third, it seems this work is constrained by a specified FPGA device, which is the Lattice ICE40UP5K. At the control algorithm introduced in this paper is based on GA and PID. As the authors propose an open architecture, can the users choose different FPGA devices and control algorithms?

As mentioned in both the conclusions and objectives, the modules implemented in the FPGA are portable to different manufacturers. Therefore, the complete solution is replicable for the interested readers.

Fourth, how the bitstream is generated? It is not clear how the control algorithm/software is converted to hardware description language and sythesized. It may be more interesting to introduce the open source synthesizer if it is developed by the authors.

Thank you for your observation. The requested information has been added in section 4. We use the project IceStorm.

I won't comment on the experimental results as they do not support the authors' original idea. At the same time, the presented architecture is quite limited targeting a specified step motor control, which a not generic.

The described architecture is generic since analogue or PWM control outputs are proposed either to use a servo amplifier or to build a driver itself. Moreover, the incorporation of additional degrees of freedom can be easily realized, as it is described in sections 4 and 5. The case study is not about the control of a stepper motor. It is focused on controlling motors for industrial use. The characteristics of the employed test bench is described in section 5.

Reviewer 4 Report

This article describes an open-source FPGA based PID controller tuned by Genetic Algorithm.

The state of the art in servo system controllers and programmable logic elements was presented.

An objective function was formulated and the methodology was described.

The English language needs extensive editing and spell check. Some of the corrections that must be done are (these are just few examples out of many):

The sentence in lines 28-29 is not clear, I suggest to revise it.

In line 28 correct the word "adn" to "and".

Headline of chapter 4 is not in English.

The sentence in lines 165-166 is not clear.

In many places through the article authors write "AG" instead of "GA".

Another major issue is regarding the use and description of GA:

First, the appropriate term is "crossover" and not "crossing". Also, revise table 1 to match the formulations described in figure 3 (i.e., I believe authors intended to write crossover operator instead of reproduction).

Second, what was the explicit GA code used for your application? How the GA parameters were calibrated? The provided general pseudocode in table 1 is too general and not satisfactory for understanding the real algorithm that was used. By looking at the pseudocode, I have firstly assumed that you have performed mutation in each iteration rather than 12% of the time as described later on. Also, I can hardly imagine how you were able to converge to the optimal result after only 20 iterations of GA with such large fluctuations of the objective function values according to figure 8.  How do you explain this fast convergence?

Authors also must indicate how many simulations were performed, and are the results were statistically significant.

What are the values of 'alpha', 'beta' and 'gama' in the objective function? and how do different values of these parameters affect the results?

Authors must write the results values in addition to their graphical representation in figures 9-11.

Author Response

REVIEWER 4

This article describes an open-source FPGA based PID controller tuned by Genetic Algorithm.

The state of the art in servo system controllers and programmable logic elements was presented.

An objective function was formulated and the methodology was described.

The English language needs extensive editing and spell check. Some of the corrections that must be done are (these are just few examples out of many):

The sentence in lines 28-29 is not clear, I suggest to revise it.

Thank you very much for  your comment. A native English speaker revised the manuscript.

In line 28 correct the word "adn" to "and".

Thank you very much for  your comment. We’ve changed the word.

Headline of chapter 4 is not in English.

Thank you very much for  your comment. A native English speaker has revised the manuscript.

The sentence in lines 165-166 is not clear.

Thank you very much for  your feedback. The sentence has been changed.

In many places through the article authors write "AG" instead of "GA".

Thank you very much for  your observations. We’ve fixed this mistake.

Another major issue is regarding the use and description of GA:

First, the appropriate term is "crossover" and not "crossing". Also, revise table 1 to match the formulations described in figure 3 (i.e., I believe authors intended to write crossover operator instead of reproduction).

We appreciate the comment. We’ve fixed this issue.

Second, what was the explicit GA code used for your application? How the GA parameters were calibrated? The provided general pseudocode in table 1 is too general and not satisfactory for understanding the real algorithm that was used. By looking at the pseudocode, I have firstly assumed that you have performed mutation in each iteration rather than 12% of the time as described later on. Also, I can hardly imagine how you were able to converge to the optimal result after only 20 iterations of GA with such large fluctuations of the objective function values according to figure 8.  How do you explain this fast convergence?

Thank you very much for your observations. We’ve re-built the pseudocode to give a better understanding for the interested readers (see section 4). In the results section the statistical analysis of the GA was added as well.

Authors also must indicate how many simulations were performed, and are the results were statistically significant.

Thank you very much for  your feedback. In this revised manuscript, we’ve indicated the simulations that were performed (see section 4). In the results section the statistical analysis of the GA was added as well.

What are the values of 'alpha', 'beta' and 'gama' in the objective function? and how do different values of these parameters affect the results?

We appreciate your kind commentary. In the revised manuscript, we have explained the terms alpha, beta and gamma and how different values of these parameters affect the results (see the lines 152-157).

Authors must write the results values in addition to their graphical representation in figures 9-11. 

We appreciate your kind commentary. The Results section has been restructured to match the paragraphs with the figures (see lines 344-365).

Round 2

Reviewer 2 Report

1) My several questions, regarding the robustness of the approach, GA trapping into local optima, readjusting the control plant, left unanswered.

2) How can a population size be set to 0? (Alg. 1).

3) Fig. 9 is missing from the x and y units (labels).

4) 1024 PPR is a very poor incremental encoder resolution.

5) A new author is included in the paper without explicitly notifying reviewers.

6) Plots are improved, but still missing from graphical elements, such as gridlines, labels, larger and more readable fonts,...

Author Response

REVIEWER 2

1) My several questions, regarding the robustness of the approach, GA trapping into local optima, readjusting the control plant, left unanswered.

The trend of each cost function is depicted in Figure 9b. The statistical analysis was performed in section 5 from lines 299-349 to highlight the robustness of the algorithm. The times it takes to reset are between 4 and 7.5 s. The focus of the article is to retrofit the proposed test bench. Then, if a motor change or modification is made to the mechanical system, a readjustment can be performed in the parameters of the initial configuration. However, it depends on the experience and knowledge of the plant which contributes greatly to the rapid convergence. When the algorithm does not converge, then the test bench presents great energy consumption and vibrations. Thus, it is possible to implement an emergency stop from the GUI for safety in any control system.

2) How can a population size be set to 0? (Alg. 1).

We are sorry for the mistake. It was a typo during the writing of the algorithm. The initial population of 10 individuals was employed.

3) Fig. 9 is missing from the x and y units (labels).

Figure 9 has been edited, and labels were added.

4) 1024 PPR is a very poor incremental encoder resolution.

The encoder resolution is predefined by the motor manufacturer, and according to the topic of the research, a retrofit (the reuse of a mechatronic system) has been carried out. Moreover, X4 encoding was employed during data acquisition. Therefore the accuracy is further increased four times (i.e., 4096 pulses)

5) A new author is included in the paper without explicitly notifying reviewers.

The co-author who was added to the article is a specialist in intelligent control and contributed to analyzing the performance of the self-tuning algorithm. He also contributed to the grammatical revision of the writing. This was notified to the editorial office in the first round.

6) Plots are improved, but still missing from graphical elements, such as gridlines, labels, larger and more readable fonts,...

Thank you for the comment. In the latest version of the manuscript, all the plots and images are vectorized (best quality for digital pictures).

Reviewer 3 Report

The authors have addressed most of my comments in the first round review process. I have one more question to the authors. In table 3, the authors use 0.1ms for the sampling time. Refer to this paper: X.Wu and R.Goodall, One-bit processing for digital control, IEE Proceedings - Control Theory and Applications, Volume: 152 , Issue: 4 , 8 July 2005, at such a high sampling frequency, there might be a numerical issue. The authors use 16-bit and 24-bit fixed point data format. Is this configuration sufficient at 0.1ms sampling rate? Can the authors provide a time analysis for the PID controller and GA algorithm. The paper can be accepted if this is clarified.

Author Response

REVIEWER 3

The authors have addressed most of my comments in the first round review process. I have one more question to the authors. In table 3, the authors use 0.1ms for the sampling time. Refer to this paper: X.Wu and R.Goodall, One-bit processing for digital control, IEE Proceedings - Control Theory and Applications, Volume: 152 , Issue: 4 , 8 July 2005, at such a high sampling frequency, there might be a numerical issue. The authors use 16-bit and 24-bit fixed point data format. Is this configuration sufficient at 0.1ms sampling rate? Can the authors provide a time analysis for the PID controller and GA algorithm. The paper can be accepted if this is clarified.

Table 1 shows the logical resources of the full system implementation. This information also presents the minimum periods (maximum frequency) of the operation of the controller. Our system has a main clock of 24MHz, the sampling time is 100 us, and the timing analyzer of the synthesis tool (synthesized hardware and hardware routing) throws the most critical path with a latency of 32 ns. Therefore, the system has no problems with performing control in real-time. However, the recommended reference is a very interesting work since it both provides a new methodology in digital controllers and focuses on minimizing resources, which is why it replaces multipliers with ‘conditional switches’. As mentioned before, multipliers are the resources that in any digital system generate latency. So it was decided to integrate the reference.

Reviewer 4 Report

Authors have answered most of my comments, and the paper has been significantly improved. Nevertheless, there are still few things that were not properly responded:

  1. Although authors stated that they have explained how different values of the terms alpha, beta and gamma affect the results (in lines 152-157), I have failed to find this explanation in the manuscript.
  2. Although authors included additional explanations to the results of figures 9-11 (or 11-13 in the revised manuscript), my comment regarding writing the result values in addition to their graphical representation in figures 9-11 was not properly addressed. Authors should add some values (in numbers) to the text.

Author Response

REVIEWER 4

Authors have answered most of my comments, and the paper has been significantly improved. Nevertheless, there are still few things that were not properly responded:

  1. Although authors stated that they have explained how different values of the terms alpha, beta and gamma affect the results (in lines 152-157), I have failed to find this explanation in the manuscript.

These values are modified through the restrictions, so when a restriction leaves the search region, the objective function is strongly penalized. If these restrictions are achieved, the values of alpha, and beta, are 1 while gamma remains at 0.

  1. Although authors included additional explanations to the results of figures 9-11 (or 11-13 in the revised manuscript), my comment regarding writing the result values in addition to their graphical representation in figures 9-11 was not properly addressed. Authors should add some values (in numbers) to the text.

We appreciate the comment. We’ve fixed this issue in the manuscript.

Round 3

Reviewer 3 Report

I am satisfied with the revision.